# Intraoperative Parathyroid Gland Identification Using Autofluorescence Imaging in Thyroid Cancer Surgery with Central Neck Dissection: Impact on Post-Operative Hypocalcemia

**DOI:** 10.3390/cancers16010182

**Published:** 2023-12-29

**Authors:** Joanne Guerlain, Ingrid Breuskin, Muriel Abbaci, Livia Lamartina, Julien Hadoux, Eric Baudin, Abir Al Ghuzlan, Sophie Moog, Alix Marhic, Adrien Villard, Rais Obongo, Dana M. Hartl

**Affiliations:** 1Department of Head and Neck Cancer and ENT Surgery, Gustave Roussy, 114 Rue Edouard Vaillant, 94805 Villejuif, France; joanne.guerlain@gustaveroussy.fr (J.G.);; 2Plate-Forme Imagerie et Cytométrie, UMS 23/3655, Gustave Roussy, Université Paris-Saclay, 114 Rue Edouard Vaillant, 94805 Villejuif, France; 3Département d’Imagerie, Service d’Oncologie Endocrinienne, Gustave Roussy, 114 Rue Edouard Vaillant, 94805 Villejuif, France; 4Department of Pathology, Gustave Roussy, 114 Rue Edouard Vaillant, 94805 Villejuif, France; 5Department of Head and Neck Cancer and ENT Surgery, Henri Becquerel Cancer Center, Rue d’Amiens CS 11516, 76038 Rouen, France; franchel-rais.obongo-anga@chb.unicancer.fr

**Keywords:** thyroid cancer, central neck dissection, parathyroid autofluorescence, near-infrared autofluorescence, post-operative hypocalcemia

## Abstract

**Simple Summary:**

This study describes the impact of near-infrared autofluorescence (NIRAF) on identifying the parathyroid glands on post-operative hypocalcemia in cancer surgery with total thyroidectomy, systematically associated with central neck dissection. In our study, the use of NIRAF significantly decreased the rate of PO hypocalcemia <2.10 mmol/L (36% vs. 60%, *p* = 0.003). Our study highlights the fact that NIRAF is a surgically non-invasive adjunct, and can improve patients’ outcomes for thyroid cancer surgery by reducing post-operative temporary hypoparathyroidism for patients, with excellent oncologic prognosis.

**Abstract:**

Hypoparathyroidism is the most frequent complication in thyroid surgery. The aim of this study was to evaluate the impact of intraoperative parathyroid gland identification, using autofluorescence imaging, on the rate of post-operative (PO) hypoparathyroidism in thyroid cancer surgery. Patients undergoing total thyroidectomy with central neck dissection from 2018 to 2022 were included. A prospective cohort of 77 patients operated on using near-infrared autofluorescence (NIRAF+) with the Fluobeam^®^ (Fluoptics, Grenoble, France) system was compared to a retrospective cohort of 94 patients (NIR−). The main outcomes were the rate of PO hypocalcemia, with three cutoffs: corrected calcium (Cac) < 2.10 mmol/L, <2.00 mmol/L and <1.875 mmol/L, and the rate of permanent hypoparathyroidism, at 12 months. The rate of PO Cac < 2.10 mmol/L was statistically lower in the NIRAF+ group, compared to the control group (36% and 60%, *p* = 0.003, respectively). No statistically significant difference was observed for the other two thresholds. There was a lower rate of permanent hypoparathyroidism in the NIRAF+ group (5% vs. 14% in the control group), although not statistically significant (*p* = 0.07). NIRAF is a surgically non-invasive adjunct, and can improve patients’ outcomes for thyroid cancer surgery by reducing post-operative temporary hypoparathyroidism. Larger prospective studies are warranted to validate our findings.

## 1. Introduction

Hypoparathyroidism is the most frequent complication in thyroid surgery. Temporary hypoparathyroidism after thyroid surgery is estimated to occur in 6.9% to 46% of patients, and the rate of permanent hypoparathyroidism is estimated to be 0% to 6.6% [1,2,3,4,5,6]. The risk of hypoparathyroidism increases with the extent of thyroid resection and the association of central neck dissection, and decreases with the expertise of the center and the experience of the surgeon [1]. It is a permanent challenge for surgeons to improve the functional outcomes after surgery, particularly in cancer thyroid surgery, which has an excellent long-term prognosis.

Permanent hypoparathyroidism after total thyroidectomy has been shown to be associated with an increased risk of cardiovascular and renal morbidities, a higher risk of malignancies, and an increased risk of mortality [7,8]. Even when effectively treated, permanent hypoparathyroidism is still associated with a decreased quality of life [9].

Parathyroid glands (PGs) are small and have a variable location, aspect and vascularization. Near-infrared autofluorescence (NIRAF) is a non-invasive technique permitting intraoperative identification of PGs by autofluorescence imaging. Infrared light- and fluorescence-based imaging technologies are new adjuncts to surgery. Infrared cameras can be used to map vascularity when a fluorophore is injected intravenously, or to map nerves by injecting nerve-specific fluorophores [10]. Autofluorescence is a relatively new technique for identifying and preserving PGs intraoperatively [10,11], approved by the FDA for clinical use since 2019 [12]. It is based on the presence of an endogenous fluorophore, yet to be identified, within the parathyroid glands themselves, which fluoresces when the parathyroid gland is illuminated with a laser light of a certain wavelength (785 nm). The fluorescent light given off by the parathyroid gland can be picked up by an infrared camera. This enables the surgeon to better see the parathyroid gland that “lights up”, as compared to the surrounding thyroid tissue, in an entirely non-invasive fashion, without the injection of dyes or exogenous fluorophores [10]. This technology has been shown, in a prospective randomized study, to improve the rate of temporary hypocalcemia in the setting of total thyroidectomy, as well as the rate of inadvertent parathyroid resection and the need for parathyroid reimplantation [13].

A major problem in the evaluation of post-operative (PO) hypoparathyroidism is the lack of a consensual definition. In most studies, PO hypoparathyroidism is defined as a PO hypocalcemia with corrected calcium (Cac) < 2 mmol/L (=8 mg/dL) [7,8,9]. Some authors have used other definitions: different Cac thresholds (1.875 mmol/L, 1.90 mmol/L, or 2.10 mmol/L with symptoms), presence of hypocalcemia symptoms, or a low PTH level [1,14,15,16]. One prospective study found a 9.1% PO hypocalcemia rate with NIRAF, as compared to 21.7% without (*p* = 0.007), using the definition of Cac < 2.00 mmol/L, and including total thyroidectomies with 15% cancers [13]. Other prospective studies have not shown significant differences using this cutoff [14,17], although in one study there was less severe hypocalcemia with NIRAF using the cutoff of <1.90 mmol/L (1.2 vs. 11.8%, *p* = 0.005) [8].

The aim of this study was to evaluate the impact of intraoperative PG identification, using autofluorescence imaging in thyroid cancer surgery, on the rate of PO hypoparathyroidism when a total thyroidectomy and an additional central neck dissection was performed, in comparison with a total thyroidectomy and central compartment neck dissection performed without this technology. The main objective was to compare the rate of PO hypocalcemia with three cutoffs: corrected calcium (Cac) < 2.10 mmol/L (8.4 mg/dL), Cac < 2.00 mmol/L (8 mg/dL) and Cac < 1.875 mmol/L (7.5 mg/dL), and the rate of permanent hypoparathyroidism at 12 months.

## 2. Materials and Methods

### 2.1. Study Design

This study was performed in a single comprehensive cancer center from 2018 to 2022.

A retrospective cohort included patients operated on from 2018 to 2020, when NIRAF was not systematically used in the department (control group). The second cohort was a retrospective cohort of prospectively collected data of patients operated on from 2020 to 2022, using NIRAF (NIRAF group), with the with a near-infrared (NIR) camera.

The study protocol was approved by our institutional review board (IRB number 2023-190).

### 2.2. Patient Selection

We included only files of patients over 16 years old undergoing total thyroidectomy (TT) associated with central neck dissection (CND) for thyroid cancer at our institution. Central neck dissection was prophylactic when there was no suspicion of pathologic lymph nodes in the central compartment on the preoperative neck ultrasound, whereas it was therapeutic when lymph node metastases were identified preoperatively. When necessary, a lateral neck dissection (LND) was added, based on the clinical and ultrasonographic pre-operative nodal status. A bilateral central neck dissection was performed for all patients preoperatively classified as cN1b. The extent of neck dissection was discussed and validated preoperatively, in all cases, by the thyroid tumor board at our institution

Total thyroidectomy without CND and thyroid lobectomy were excluded from the study.

Patients with preoperative parathyroid disease or abnormal preoperative serum calcium levels (obtained systematically) were excluded.

### 2.3. Imaging System

The NIR camera used in the study is Fluobeam 800^®^ (Fluoptics, Grenoble, France). The hand-held optical head excites tissues with a wavelength of 750 nm at a working distance of 20 cm (5 mW/cm^2^), and collects in autofocus the fluorescence signal from the surgical field between 800 nm and 900 nm. The fluorescence videos are displayed in grey values in real-time and in two dimensions, using the Fluosoft (Fluoptics, Grenoble, France). Data can be saved in video or image format. NIRAF acquisition requires operating lights to be dimmed or turned off.

### 2.4. Surgical Technique and Post-Operative Monitoring

Central neck dissection was performed as described in a previous publication by our group, and in accordance with international guidelines [18,19]. The NIRAF-assisted cases were performed by surgeons with experience using the technology (authors JG, IB, DMH). In the NIRAF group, PGs were identified in situ or on the operating specimen after resection, using the NIR camera: during the thyroidectomy and paratracheal neck dissection, the surgeon attempted to visualize the parathyroid glands in situ by visual examination, and then with the NIR camera. If all four PGs were not visualized in situ, the resected specimen was explored ex vivo, immediately after resection, by visual examination and then with the NIR camera

Tissue suspected to be a PG ex vivo, based on autofluorescence signal acquisition, was analyzed by sending a 1–2 mm fragment to pathology for frozen section analysis, to confirm the diagnosis of parathyroid tissue before reimplantation. Once confirmed, non-cancerous parathyroid tissue was then fragmented and auto-transplanted into the sternocleidomastoid muscle.

In the control group, the standard procedure of identification of PGs was performed with visual identification by the surgeon in the operating field and on the pathological specimen. Usually, a frozen section procedure was also performed, to confirm the parathyroid tissue before reimplantation.

Serum calcium and albumin levels were measured on the morning of post-operative days 1 and 2. Calcium and alphacalcidol were administered, according to a local protocol adapted from Genser et al. and Stack et al. [20,21]. All patients had calcium supplementation at discharge, in accordance with published guidelines [22]. Patients with corrected calcium levels < 2.00 mmol/L had 1 µg of alphacalcidol in addition to calcium, with weekly testing thereafter. Patients with lower levels of calcemia received 2 to 3 µg of alphacalcidol, with weekly testing and a gradual decrease in doses.

### 2.5. Outcomes Measured and Endpoints

Patients’ characteristics were collected: age, gender, histological type of the tumor, TNM stage, background of cervical radiotherapy or thyroiditis. The details of the surgery type were collected, such as unilateral or bilateral CND, prophylactic or therapeutic CND, associated LND, and duration of the surgery.

Operative reports were examined to determine the number of PGs visualized, preserved in situ, or reimplanted.

For the NIRAF group, prospective data were collected on the number of PGs seen without and with the NIR camera, and the supplementary time estimated by the surgeon, using the NIRAF procedure.

For the two groups, the number of PGs found on the pathological reports was collected, as well as the Cac rate from biological analysis during hospitalization in the two days after the surgery and supplementary treatments at 12 months of surgery from consultations’ reports.

Different thresholds of PO hypocalcemia were defined, with three cutoffs: Cac < 2.10 mmol/L (8.4 mg/dL), Cac < 2.00 mmol/L (8 mg/dL), and Cac < 1.875 mmol/L (7.5 mg/dL).

Patients requiring emergency calcium supplementation before the serum calcemia could be measured were included in the low post-operative calcemia group with Cac < 1.875 mmol/L (7.5 mg/dL).

Permanent hypoparathyroidism was defined as needing supplementation twelve months after surgery.

### 2.6. Statistical Analysis

Groups of patients were compared using a Fischer’s exact test (BiostaTGV software program, https://biostatgv.sentiweb.fr/, accessed on 24 December 2023).

Significance level was defined at *p* < 0.05.

## 3. Results

### 3.1. Patient and Tumor Characteristics

Between 2018 and 2020, 94 consecutive patients underwent a TT with CND, and were included in the control group.

Between 2020 and 2022, 77 patients were included in the NIRAF group.

The patient and tumor characteristics are shown in Table 1.

### 3.2. Number of Identified PGs

A total of 253 PGs were identified only by visual examination in the 94 patients of the control group, compared to 235 in the 77 patients of the NIRAF group (visual examination completed by autofluorescence imaging, Figure 1).

There was no statistically significant difference between the two groups regarding the number of PGs identified, the number of inadvertently resected Ps in the pathological reports, or the rate of PG autotransplantation (Table 2).

In the NIRAF group, a PG was highlighted ex vivo with the help of an NIR camera in 12/77 cases (16% of the cases), permitting an autotransplantation (Figure 2).

### 3.3. PO Hypocalcemia (Table 3)

The rate of PO Cac < 2.10 mmol/L was significantly lower in the NIRAF group, compared to the control group: 36% vs. 60%, respectively (*p* = 0.003).

No statistically significant difference was observed for the other thresholds: 19% in the NIRAF group had a PO Cac < 2.00 mmol/L, compared to 27% in the control group (*p* = 0.36). The rate of PO Cac < 1.875 mmol/L was 3% in the NIRAF group, compared with 11% in the control group (*p* = 0.07).

Three patients required emergency calcium supplementation before the serum calcemia could be measured in the control group, and two in the NIRAF group. These patients were included in the low post-operative calcemia group with Cac < 1.875 mmol/L.

There was an evidently lower rate of permanent hypoparathyroidism in the NIRAF+ group (5% vs. 14% in the control group), although not statistically significant (*p* = 0.07).

**Table 3 cancers-16-00182-t003:** Comparison of groups, according to different definitions of hypocalcemia.

Calcemia mmol/L	Control Group	NIRAF Group	OR/CI (95%)	*p*-Value
Temporary <2.10	56/94 (60%)	28/77 (36%)	OR: 2.5642 CI (95%): [1.3258; 5.0351]	0.003
Temporary <2.00	25/94 (27%)	15/77 (19%)	OR: 1.4941 CI (95%): [0.6856; 3.345]	0.36
Temporary <1.875	10/94 (11%)	2/77 (3%)	OR: 1.962 CI (95%): [0.6525; 6.6501]	0.07
Permanent	13/90 (14%) *	6/76 (5%) **	OR: 3.0203 CI (95%): 0.88; 13.3038]	0.07

OR: Odds Ratio; CI (95%): 95% Confidence Interval; * Missing data in 4 patients; ** Missing data in 1 patient.

### 3.4. Supplementary Time Using NIRAF Procedure

According to the surgeons’ subjective appreciation, the NIRAF procedure took less than 10 min in 23/77 cases (30%), between 10 and 20 min in 33/77 cases (43%), between 20 and 30 min in 10/77 cases (13%), and more than 30 min in 2/77 cases (3%). Data were missing for 9/77 cases.

There was no significant difference, however, in the total objective duration of surgery between the two groups (Table 1).

## 4. Discussion

According to the results of our study, the use of the NIRAF procedure may reduce the rate of mild PO Cac, and a trend towards lower rates of permanent hypoparathyroidism was observed. This technique appears easily applicable, and did not impact the effective duration of the surgical procedure.

The definition of PO hypoparathyroidism is not consensual in the literature. Depending on the definition used, the percentage of PO hypoparathyroidism can be very different. Furthermore, there is currently no consensus regarding the importance of monitoring serum parathormone levels rather than calcemia; in practice, it is the hypocalcemia that may cause symptoms post-operatively, but in long-term studies it is still unclear how chronic hypoparathyroidism leads to increased morbidity [7,8]. Previous studies have used varying cutoffs for defining hypocalcemia, and it is still not clear if dosing the corrected calcemia is as sensitive as ionized calcium for measuring post-operative hypoparathyroidism. In our study, different cutoffs of hypocalcemia were used to define the PO hypoparathyroidism. The use of NIRAF significantly decreased the rate of PO hypocalcemia by < 2.10 mmol/L (36% vs. 60%, *p* = 0.003). This threshold is higher than the usual definition of PO hypocalcemia in the literature, in which the cutoff of <2.00 mmol/L is more frequently observed.

When we used the other cutoffs, the percentages of PO hypocalcemia were lower with the use of NIRAF, but not significantly: 19% vs. 27% for the Cac < 2.00 mmol/L (*p* = 0.36) and 3% vs. 11% for Cac < 1.875 mmol/L (*p* = 0.07). This may be due to a lack of statistical power with an insufficient number of patients, and larger prospective controlled studies are warranted. Unfortunately, we did not have sufficient data for all patients in this retrospective study to evaluate post-operative parathormone levels, which would be a more specific and sensitive means of evaluating the autofluorescence technique.

To our knowledge, only one other retrospective study investigating PG identification using autofluorescence imaging with TT + CND [16] has been published, to date. Kim et al. [16] described significantly less PO hypoparathyroidism, defined as a PTH level < 15 pg/mL with NIRAF (33.7% vs. 46.6% *p* = 0.002). There was no difference between the NIRAF group and the control group, however, when they analyzed post-operative ionized calcium levels during hospitalization, one month, 3 months and 6 months post-operatively. These results suggest, once again, the difficulties pertaining to the definition of PO hypoparathyroidism and the need for a consensual definition in order to compare the results among studies.

Discrepancies between studies may be due, in part, to surgeon experience and to the learning curve using the technology, but also due to differing patients’ parathyroid anatomy and the effect of different thyroid pathologies on parathyroid anatomy and vascularization. Preoperative serum vitamin D levels may also come into effect when evaluating the rate of hypocalcemia. In our study, only mild hypocalcemia seemed to be affected by the use of the NIRAF technology. This is difficult to explain, given that the two patient groups were comparable in terms of total thyroidectomy and central neck dissection. Our cohort may be too small to show differences at other cutoffs. Our patients routinely receive preoperative vitamin D supplementation. Directly dosing parathormone and/or ionized calcium may have given different results, but these data were not available, due to the retrospective nature of our study.

In our study, consistent with other studies in the literature, there was no significant difference in permanent hypoparathyroidism with NIRAF [13,14,16,17]: 5% with NIRAF vs. 14% in the control group, *p* = 0.07. Our rate of permanent hypoparathyroidism, as defined clinically by calcium and/or vitamin D supplementation at 12 months, was higher than rates in previously published studies, ranging from 0% to 3% [6,23]. Even in the study by Kim et al. [16], studying patients treated with central neck dissection, the incidence of permanent hypoparathyroidism as defined by a level of ionized serum calcium < 1.09 mmol/L was only 1.1% in both study groups, although there was no information as to the number of patients receiving supplementation post-operatively. Our rates of 5% and 14%, respectively, are closer in magnitude to other studies in thyroid cancer surgery with CND, with rates of permanent hypoparathyroidism from 3.6% to 44% [24,25,26,27,28]. In the study by Salem et al. [24], the rate of post-operative therapy after TT + CND ranged from 6.6% to 17.1%, depending on the definition of the therapy: oral calcium 6 months after surgery (17.1%), oral active vitamin D therapy 6 months after surgery (8.3%), or oral active vitamin D and calcium therapy 6 months after surgery (6.6%). Our numbers may have overestimated the rate of permanent hypoparathyroidism, with patients staying on therapy longer than truly necessary. Moreover, the selection of our patients could also represent a bias, as our comprehensive cancer center is a referral center and more than half of the procedures were therapeutic central neck dissection. An evaluation of the PTH levels would maybe have provided a better estimate of the true rate of permanent clinical hypoparathyroidism.

In some studies, the use of NIRAF significantly increased the number of PG visualized during surgery [29,30]. In our study, there was an increase in the number of PG visualized in the NIRAF group, even if the difference was not significant (*p* = 0.056). There was no statistically significant difference in the rate of autotransplantation or in the number of PGs in the pathological reports. Nevertheless, a PG was identified ex vivo using the autofluorescence system in 12/77 cases (16% of the cases), permitting an autotransplantation.

Identifying PGs has been shown to be correlated with a reduction in PO hypocalcemia [6]. Bergenfelz et al. [31] showed that fewer PGs identified during surgery was associated with a higher rate of transient hypocalcemia, independent of the extent of thyroidectomy or neck dissection. Tomusch et al. [5] showed that the identification of fewer than two PGs during surgery was associated with permanent hyopocalcemia, independently of the extent of thyroidectomy or the surgical indication. Other studies, however, have failed to correlate the number of PGs visualized with transient post-operative hypocalcemia [6]. This could be explained by the risk of devascularization of the PG during the dissection, but also by different definitions and cutoffs used.

Discordances between studies using NIRAF on the incidence of PO hypocalcemia and the lack of statistical significance in several studies may be explained by several hypotheses. First, most studies are retrospective, or small prospective studies, and lack the statistical power to prove the impact of NIRAF on PO hypoparathyroidism [13,14,16,17]. Moreover, studies are mainly conducted by high-volume, experienced thyroid surgeons, with experience in identifying parathyroid glands. NIRAF may show a higher rate of improvement in outcomes for low-volume surgeons. Finally, NIRAF does not evaluate the vascular perfusion of the PG. Some studies have evaluated NIRAF associated with indocyanine green (ICG), in an attempt to reduce devascularization [32]. For now, the use of ICG angiography for preservation of the parathyroid vasculature has not been well codified. A large multicenter phase 2/3 study is currently including patients to detect the parathyroid glands using NIRAF associated with ICG (Balasubramian et al., Sheffield, UK, IRAS number 287123, ISRCTN 59074092) [33].

NIRAF has some limits. In fact, there are false-positive cases [11,34], such as brown fat, colloid nodules or metastatic lymph nodes. In cancer surgery, we prefer to perform frozen section analysis on a millimeter-sized fragment on the parathyroid before reimplantation, so as not to inadvertently reimplant cancerous tissue.

In our study, 73% of the procedures with NIRAF were declared by the surgeon as taking less than 20 supplementary minutes, but there was no difference in the total objectively measured duration of surgery between the NIRAF group and the control group. More lateral neck dissections were performed in the control group, however, which may have led to a bias in the comparison of the duration of surgery.

## 5. Conclusions

In conclusion, NIRAF may be a helpful non-invasive tool to improve post-operative outcomes in patients treated for thyroid cancer. Further studies are needed to better evaluate the technology, particularly with regard to ICG, and there is a pressing need for a consensus in terms of the definition of post-operative hypoparathyroidism.

## Figures and Tables

**Figure 1 cancers-16-00182-f001:**
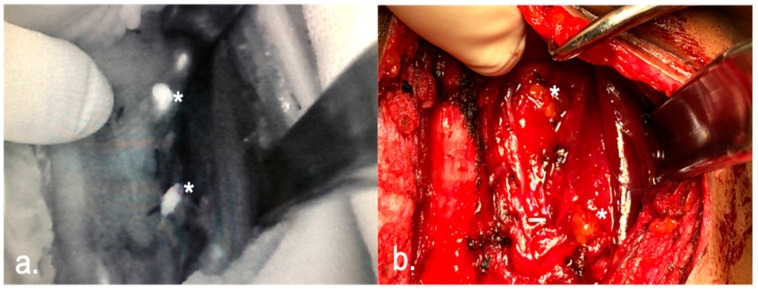
Two PGs (*) visualized with NIRAF (**a**), and without (**b**), after TT.

**Figure 2 cancers-16-00182-f002:**
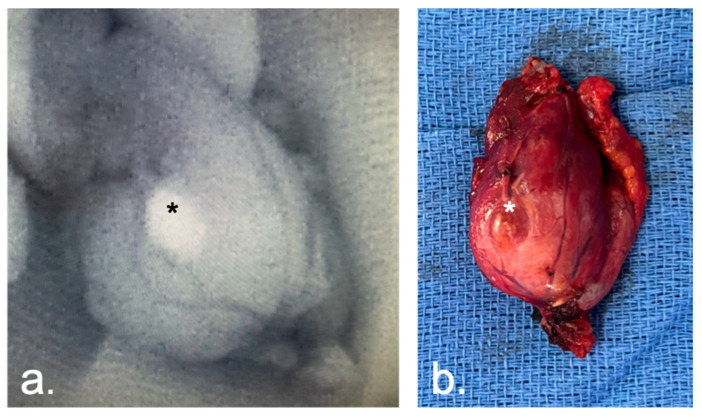
PG (*) identified on the specimen after lobectomy with NIRAF (**a**), and without (**b**).

**Table 1 cancers-16-00182-t001:** Characteristics of patients.

	Patients, No. (%)		
	Control Group (n = 94)	NIRAF Group (n = 77)	*p*-Value
Sex			0.4
Female	60 (64)	54 (70)	
Male	34 (36)	23 (30)	
Age mean, (min–max), y	50 (16–85)	51 (19–79)	0.7
Backgroud			
Radiotherapy in childhood	2 (2)	1 (1)	1
Thyroiditis	20 (21)	17 (22)	1
Histological type			0.6
Papillary carcinoma (PC)	83 (88.5)	66 (86)	
Medullary carcinoma (MC)	5 (5.5)	9 (12)	
Both PC + MC	2 (2)	1 (1)	
Poorly diff carcinoma	2 (2)	0 (0)	
Others *	2 (2)	1 (1)	
pT stage			**0.005**
T1	46 (49)	52 (68)	
T2	23 (24)	19 (25)	
T3	21 (22)	2 (3)	
T4	2 (2)	2 (3)	
T0/Tx **	2 (2)	2 (3)	
Multifocal	45 (48)	34 (44)	0.4
pN stage			0.3
N0	30 (32)	29 (38)	
N1a	13 (14)	15 (19)	
N1b	51 (54)	33 (43)	
N+ in CND			0.3
Yes	58 (62)	41 (53)	
No	36 (38)	36 (47)	
ATA 2015 risk stratification system			0.5
Low risk	10 (11)	13 (17)	
Intermediate risk	55 (59)	37 (48)	
High risk	19 (20)	16 (21)	
Not applicable (not a PC)	10 (11)	11 (14)	
Type of central neck dissection			0.7
Unilateral prophylactic CND	2 (2)	2 (3)	
Bilateral prophylactic CND	45 (48)	39 (51)	
Unilateral therapeutic CND	4 (4)	1 (1)	
Bilateral therapeutic CND	43 (46)	35 (45)	
Associated lateral neck dissection	88 (85)	51 (66)	**<0.001**
Duration of surgery, mean (min–max), minutes	161 (63–414)	158 (80–420)	0.8

* Others: 2 Non-invasive follicular thyroid neoplasm (NIFTP) with papillary-like nuclear features in the control group and a lipoadenoma with bizarre nuclei in the NIRAF group. ** Tx. In the control group, two T0s had pre-operative Bethesda VI cytology and were histological NIFTP. In the NIRAF group, one Tx was thyroiditis with fibrosis on the TT specimen, but was N1b for papillary carcinoma. The other one was a T0 Bethesda VI, and was finally, on the histological report, a lipoadenoma with bizarre nuclei.

**Table 2 cancers-16-00182-t002:** Identified PGs in NIRAF and control group.

		Patients, No. (%)		
		Control Group	NIRAF Group	*p*-Value
Number of PGs visualized in operating room				
Total		253/364 (69%)	235/308 (76%)	0.056
Mean		2.8	3	
Median		3	3	
Number of PGs vizualized per case	0	0/91 (0%)	0/77 (0%)	0.2
	1	6/91 (7%)	3/77 (4%)	
	2	30/91 (33%)	19/77 (25%)	
	3	33/91 (36%)	26/77 (34%)	
	4	22/91 (24%)	29/77 (38%)	
		(3 missing data)		
Number of patients with PG reported on pathological report		47/94 (50%)	32/77 (42%)	0.3
Rate of autotransplantation				
Number of cases		57/92 (62%)(2 missing data)	52/77 (68%)	0.5
Number of PGs		74/368 (20%)	77/308 (25%)	0.1

## Data Availability

The data presented in this study are available in this article.

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
