# Peer review of "Intraoperative Parathyroid Gland Identification Using Autofluorescence Imaging in Thyroid Cancer Surgery with Central Neck Dissection: Impact on Post-Operative Hypocalcemia"

_cancers, 2023, doi:10.3390/cancers16010182_

Round 1

Reviewer 1 Report

Comments and Suggestions for Authors

This is an interesting and well done study dealing with a significant topic. 

The technique described in this study may contribute to preserve functional parathyroid glands in order to reduce the rate of permanent hypoparathyroidism (HP), a severe complication after thyroid surgery.  According to the Authors, we need a consensual definition of HP to better compare the results of the studies.

Questions to ask Authors:

1) why so many "associated lateral neck dissections" in your clinical records? 

2) considering the high number of bilateral prophylactic CND reported in your series, do you think is it admissible to perform only unilateral CND in early stages of DTT in order to reduce the risk of hypoparathyroidism? The literature data on this topic are very heterogeneous. 

Author Response

We thank the authors for their time and effort in reviewing our work and for their interesting and pertinent comments that will improve our manuscript.

Reveiwer 1

This is an interesting and well done study dealing with a significant topic. 

The technique described in this study may contribute to preserve functional parathyroid glands in order to reduce the rate of permanent hypoparathyroidism (HP), a severe complication after thyroid surgery.  According to the Authors, we need a consensual definition of HP to better compare the results of the studies.

Questions to ask Authors:

  • why so many "associated lateral neck dissections" in your clinical records? 

Response : our center is a cancer center that receives many cases of patients with clincally detectable lymph node metastases, and who require therapeutic neck dissection. Central neck dissection is often associated with lateral neck dissection, and so our cohort is enriched with patients classified not only cN1a but also cNb.

2) considering the high number of bilateral prophylactic CND reported in your series, do you think is it admissible to perform only unilateral CND in early stages of DTT in order to reduce the risk of hypoparathyroidism? The literature data on this topic are very heterogeneous. 

Response : Yes, we only perform prophylactic neck dissection now for high-risk patients or in the context of a prospective randomized clinical trial that is currently enrolling. We do agree that for low risk cancers, unilateral or no neck dissection is most probably oncologically sound and functionally more appropriate.

Reviewer 2 Report

Comments and Suggestions for Authors

Thanks to opportunity to review this interesting paper. You report an important experience on a topic of thyroid surgery.

I think that there are some importantant bias.

The 2 group are not compareble. You should exclude the patients with lateral neck dissection, more frequent in control group, in which we have a higest percentage of hypocalcemia.

The number of Pg identified, resecet, o with autotransplantation in the same in the 2 groups.

Why have you a different results? Why do you think  that the results depend of use of Probe?

You a personal evaluetion of  transient hypocalcemia.

The value of calcium < 2,10mm/L is really hypocalcemia? How many patients needde a terapy?

Did you changed your specific treatment for these patients?

Really , I think that , with the results of your study, that the use of Probe dosen't help to reduce the hypocalcemia post thyroidectomy.

Author Response

We thank the authors for their time and effort in reviewing our work and for their interesting and pertinent comments that will improve our manuscript.

Reviewer 2

Thanks to opportunity to review this interesting paper. You report an important experience on a topic of thyroid surgery.

I think that there are some importantant bias.

The 2 group are not compareble. You should exclude the patients with lateral neck dissection, more frequent in control group, in which we have a higest percentage of hypocalcemia.

Response : Indeed, in table 1 we show that the T-stage was different between the groups, but in the same table we show that the extent of neck dissection and the ATA (American Thyroid Association) risk groups are comparable in our 2 patient cohorts. Lateral neck dissection does not influence the rate of hypocalcemia, only central neck dissection has been shown to affect parathyroid gland function, so including patients with a lateral neck dissection should not, theoretically, affect the outcomes we studied.

The number of Pg identified, resecet, o with autotransplantation in the same in the 2 groups.

Why have you a different results? Why do you think  that the results depend of use of Probe?

Response: There was a trend (p=0.056) in seeing more parathyroid glands intraoperatively in the NIRAF group. The explanation may be that our cohort is too small for significant statistical analysis. We do think that, similar to neuromonitoring in thyroid surgery, the use of NIRAF makes us more aware of the problem and more careful to look for and to preserve or reimplant parathyroids.

You a personal evaluetion of  transient hypocalcemia. The value of calcium < 2,10mm/L is really hypocalcemia? How many patients needde a terapy? Did you changed your specific treatment for these patients?

Response: We agree with the reveiwer that this level is almost normal, and we really don’t know why calcemia was only different at this level and not at lower levels. We have added this remark in the Discussion. Most of our patients were discharged with calcium supplementation +/- alfacalcidol as a preventive measure for symptoms. This has been added to the Patients and Methods section.

It would probably be more sensitive and specific to measure postoperative serum parathormone levels, but unfortunately, due to the retrospective nature of our study, we did not have this data for a sufficient number of patients.

Really , I think that , with the results of your study, that the use of Probe dosen't help to reduce the hypocalcemia post thyroidectomy.

We agree with the reviewer that the results of our study do not seem to show a very significant difference between our patient groups. Some published studies have also shown « negative » results, whereas others show a more convincing difference. This probably is due to study design and patient populations. We hope to see more prospective randomized studies in order to have a better level of evidence one way or the other.

Reviewer 3 Report

Comments and Suggestions for Authors

Introduction:

- Provide more background on the anatomy and blood supply of the parathyroid glands and their role in calcium regulation. cite PMID:31646536

- Discuss the prevalence of postoperative hypocalcemia after thyroid surgery, both temporary and permanent. Expand on associated morbidity.

- Explain the challenges of visually identifying all parathyroid glands during thyroidectomy and central neck dissection. discuss exoscope and cite  doi:10.3390/jcm11133639

- Introduce near infrared autofluorescence (NIRAF) technology in more detail - mechanism, prior applications in parathyroid imaging. cite doi:10.3760/cma.j.cn112144-20230817-00089.

- Discuss previous studies evaluating NIRAF for parathyroid identification during thyroid surgery and their limitations.

- Clearly state the rationale and objectives for the current study.

Methods:

- Expand on patient eligibility - inclusion and exclusion criteria. Provide details on indication for central neck dissection.

- Describe the surgical procedure for thyroidectomy and central neck dissection in both groups. Standardize techniques between groups.

- Explain how postoperative calcium was monitored - timing and frequency of lab draws.

- Discuss supplemental calcium administration if levels dropped - oral vs IV, dosing, triggers for treatment.

- Provide more details on NIRAF system - manufacturer, excitation wavelengths, imaging modes.

- Explain how NIRAF was utilized intraoperatively - timing, verification of parathyroid tissue, impact on surgical technique.

- State specific statistical tests used to compare outcomes between groups - chi-square, t-tests etc. Specify statistical significance levels

Results:

- Present baseline characteristics of each group in a table - age, gender, tumor histology, stage, etc. Perform statistical comparisons.

- Report mean or median postoperative calcium levels with standard deviations for each group at various timepoints.

- Include 95% confidence intervals and p-values for the rates of temporary and permanent hypocalcemia.

- Provide a figure showing trends in calcium levels over follow-up time by group.

- Report the number and percentage of parathyroid glands identified by NIRAF vs visually.

Discussion:

- Discuss the challenges in defining postoperative hypoparathyroidism - optimal calcium cutoff, timing, symptoms vs labs.

- Compare results to other studies evaluating NIRAF for parathyroid identification and preservation during thyroidectomy.

- Analyze potential reasons for discrepancies in impact on hypocalcemia rates between studies. cite PMC10403213.

- Discuss possible explanations for reduced mild hypocalcemia with NIRAF but not more severe levels.

- Explore possible confounders that could have influenced results - surgical experience, case complexity, concomitant lateral neck dissections.

- Analyze strengths and weaknesses of the study design - retrospective, single center vs generalizability of findings.

- Suggest future research directions - prospective validation, combining NIRAF with novel perfusion imaging techniques.

Comments on the Quality of English Language

no

Author Response

We thank the reviewers for their time and effort in reviewing our work and for their interesting and pertinent comments that will improve our manuscript.

Reviewer 3

Introduction:

- Provide more background on the anatomy and blood supply of the parathyroid glands and their role in calcium regulation. cite PMID:31646536

Response : We think that this information is not really the subject of our study ; our readers are generally surgeons who have this background. The article cited by the reviewer is from 1958, and has been the basis for clinical practice for decades.

- Discuss the prevalence of postoperative hypocalcemia after thyroid surgery, both temporary and permanent. Expand on associated morbidity.

Response : On page 2 we have added data and 3 references (references 7, 8 and 9).

- Explain the challenges of visually identifying all parathyroid glands during thyroidectomy and central neck dissection. discuss exoscope and cite  doi:10.3390/jcm11133639

We have touched on these difficulties in the introduction and discussion. The Exoscope is actually a camera that aids in magnification. We use loupes for parathyroid dissection, which have essentially the same effect of magnification.

- Introduce near infrared autofluorescence (NIRAF) technology in more detail - mechanism, prior applications in parathyroid imaging. cite doi:10.3760/cma.j.cn112144-20230817-00089.

Response : We described the technology on page 2. The article cited above concerns infrared thermal imaging, which, while using an infrared camera, is different from parathyroid autofluorescence which requires  laser to elicit the fluorescence naturally inherent in parathyroid glands that can then be detected with an infrared camera.

- Discuss previous studies evaluating NIRAF for parathyroid identification during thyroid surgery and their limitations.

Reponse : We discuss other studies similar to ours in the Discussion, and the main limitation in comparing studies, which is the choice of definition of hypoparathyroidism and hypocalcemia.

- Clearly state the rationale and objectives for the current study.

Response : we have made modifications to the introduction accordingly.

Methods:

- Expand on patient eligibility - inclusion and exclusion criteria. Provide details on indication for central neck dissection.

            Response : We have added more information on page 3.

- Describe the surgical procedure for thyroidectomy and central neck dissection in both groups. Standardize techniques between groups.

Response : We have added information concerning our technique on page 3 and have added 2 references (references 18 and 19).

- Explain how postoperative calcium was monitored - timing and frequency of lab draws.

Reponse : we have added this information on page 4.

- Discuss supplemental calcium administration if levels dropped - oral vs IV, dosing, triggers for treatment.

Reponse : we have added this information on page 4.

- Provide more details on NIRAF system - manufacturer, excitation wavelengths, imaging modes.

Reponse : we have added this information on page 3.

- Explain how NIRAF was utilized intraoperatively - timing, verification of parathyroid tissue, impact on surgical technique.

Response :This appears on page 4.

- State specific statistical tests used to compare outcomes between groups - chi-square, t-tests etc. Specify statistical significance levels

Response :This appears on page 4.

Results:

- Present baseline characteristics of each group in a table - age, gender, tumor histology, stage, etc. Perform statistical comparisons.

Reponse : This information is presented in table 1.

- Report mean or median postoperative calcium levels with standard deviations for each group at various timepoints.

Response : Unfortunately we do not have longitudinal data (patients were discharged between day 1 and day 3, due to neck dissections). Calcemia was then monitored outside of our institution at various intervals, so that we do not have homogeneous or complete data other than the levels tested routinely postoperatively.

- Include 95% confidence intervals and p-values for the rates of temporary and permanent hypocalcemia.

Response : We have added table 3 to show these results.

- Provide a figure showing trends in calcium levels over follow-up time by group.

Response : as stated above, we do not have complete or comparable longitudinal data for our patients in this retrospective study.

- Report the number and percentage of parathyroid glands identified by NIRAF vs visually.

Response : this data is shown in table 2.

Discussion:

- Discuss the challenges in defining postoperative hypoparathyroidism - optimal calcium cutoff, timing, symptoms vs labs.

Response : we have augmented the discussion on page 8 to this effect.

- Compare results to other studies evaluating NIRAF for parathyroid identification and preservation during thyroidectomy.

Response : this part of the discussion have been augmented as requested and can be found on pages 8 and 9.

- Analyze potential reasons for discrepancies in impact on hypocalcemia rates between studies. cite PMC10403213.

Response : We have augmented the discussion as requested, on page 8. Unfortunately, we were unable to find the reference cited above.

- Discuss possible explanations for reduced mild hypocalcemia with NIRAF but not more severe levels.

Response : We have added to the discussion on this topic, as recommended by the reviewer, on pages 8 and 9.

- Explore possible confounders that could have influenced results - surgical experience, case complexity, concomitant lateral neck dissections.

Response : We have added to the discussion on this topic, as recommended by the reviewer, on pages 8 and 9. Lateral neck dissection has not been shown to affect postoperative calcium levels, but was associated with central neck in patients with clinically pathologic lymph nodes classfied as cN1b.

- Analyze strengths and weaknesses of the study design - retrospective, single center vs generalizability of findings.

Response : We have added to the discussion on this topic, as recommended by the reviewer, on pages 8 and 9

- Suggest future research directions - prospective validation, combining NIRAF with novel perfusion imaging techniques.

Response : We have added to the discussion on this topic, as recommended by the reviewer, on pages 8 and 9

Round 2

Reviewer 2 Report

Comments and Suggestions for Authors

Dear Authors, thank you for your pronty and comprensive answer.

I think the the paper is more clear in the results